# Targeting Innate Immunity in Glioma Therapy

**DOI:** 10.3390/ijms25020947

**Published:** 2024-01-12

**Authors:** Andrew G. Gillard, Dong Ho Shin, Lethan A. Hampton, Andres Lopez-Rivas, Akhila Parthasarathy, Juan Fueyo, Candelaria Gomez-Manzano

**Affiliations:** 1Department of Neuro-Oncology, The University of Texas MD Anderson Cancer Center, Houston, TX 77030, USA; aggillard@mdanderson.org (A.G.G.); dshin4@mdanderson.org (D.H.S.); lahampton@mdanderson.org (L.A.H.); arlopez3@mdanderson.org (A.L.-R.); aparthasarathy1@mdanderson.org (A.P.); 2MD Anderson Cancer Center UTHealth Houston Graduate School of Biomedical Sciences, Houston, TX 77030, USA

**Keywords:** innate immunity, glioma, immunotherapy, adaptive therapy, virotherapy

## Abstract

Currently, there is a lack of effective therapies for the majority of glioblastomas (GBMs), the most common and malignant primary brain tumor. While immunotherapies have shown promise in treating various types of cancers, they have had limited success in improving the overall survival of GBM patients. Therefore, advancing GBM treatment requires a deeper understanding of the molecular and cellular mechanisms that cause resistance to immunotherapy. Further insights into the innate immune response are crucial for developing more potent treatments for brain tumors. Our review provides a brief overview of innate immunity. In addition, we provide a discussion of current therapies aimed at boosting the innate immunity in gliomas. These approaches encompass strategies to activate Toll-like receptors, induce stress responses, enhance the innate immune response, leverage interferon type-I therapy, therapeutic antibodies, immune checkpoint antibodies, natural killer (NK) cells, and oncolytic virotherapy, and manipulate the microbiome. Both preclinical and clinical studies indicate that a better understanding of the mechanisms governing the innate immune response in GBM could enhance immunotherapy and reinforce the effects of chemotherapy and radiotherapy. Consequently, a more comprehensive understanding of the innate immune response against cancer should lead to better prognoses and increased overall survival for GBM patients.

## 1. Introduction

Primary malignant brain tumor incidence is approximately 7 per 100,000 individuals, and approximately 50% of those are glioblastomas (GBM). The standard of care for GBM has not changed since 2005 and involves surgical resection, radiotherapy, and chemotherapy, including temozolomide [1]. GBM diagnosis is associated with a median survival of approximately 15 months and a 5-year survival of approximately 36% [2,3]. These tumors are known for early infiltration, limiting the effectiveness of surgical resection [4]. Another critical challenge during GBM treatment is the heterogeneity of cancer cells in terms of phenotype, genotype, and function; therefore, attempts to identify molecular mechanisms common to all tumors have been unsuccessful. GBM, the most common and invasive primary malignancy of the central nervous system, is considered a “desert” from an immunological point of view. In other words, its tumor microenvironment is known to be “immune cold”, illustrated by a lack of tumor-infiltrating T-cells, which is probably responsible for the lack of efficacy of immunotherapy [5,6]. GBM is known to modulate and exploit normal brain cells to support the growth of transformed cells, affecting all cell types in the tumor microenvironment and inhibiting both innate and adaptive immune cell responses [7]. Novel therapies for these tumors are currently being explored, particularly those that can modulate an effective immune response and prompt lymphocyte infiltration to the tumor microenvironment. In this review, we highlight the role of innate immune activation in the context of cancer therapies, including GBMs.

The immune system is a complex network designed to protect the body against various threats, including infections and abnormal cells like cancer cells. Within this system, both innate and adaptive immunity work collaboratively to provide a multi-layered defense mechanism (Figure 1). Central to the function of the innate immune system are signaling molecules known as PAMPs (pathogen-associated molecular patterns) and DAMPs (damage-associated molecular patterns) [8]. PAMPs are essentially molecular signatures specific to pathogens such as bacteria, viruses, and fungi. These patterns are recognized by innate immune cells through specific receptors, such as the Toll-like receptors (TLRs). When these receptors detect PAMPs, they trigger an immediate response, activating the innate immune system to fight the invading pathogens. On the other hand, DAMPs are molecules released or exposed by damaged or dying cells, signaling distress or danger within the body. They can be caused by trauma, inflammation, or even cancerous changes. DAMPs act as danger signals, alerting the immune system to potential threats within the body. The innate immune system’s recognition of these PAMPs and DAMPs is critical in initiating a rapid and nonspecific response [9,10]. This initial response includes the recruitment of immune cells to the site of infection or damage, the activation of phagocytosis to engulf and eliminate threats, and the release of cytokines to coordinate the immune response. These PAMPS and DAMPS also act as cues for the activation of inflammasome signaling pathways, resulting in the production of proinflammatory cytokines such as type I interferons (IFNs). Cytokines then function to prime the adaptive immune response, mediating pathogen clearance [11,12].

Although some of the roles of innate immunity have been well studied, recent publications have suggested new aspects of innate immunity in both the initiation and advancement of tumorigenesis. For instance, abnormal cell proliferation and stress associated with carcinogenesis have been shown to upregulate the release of DAMPS [13], subsequently activating innate immune pathways to clear transforming cells [14]. This feedback loop, in turn, facilitates the recruitment, activation, and clonal expansion of tumor-specific CD8+ T-cells. These changes may promote response to therapy and better patient outcomes [15,16,17]. During this process, cells with weak immunogenic signatures are spared by the adaptive immune system and can form tumors [18,19]. Consequently, based on these observations, there has been interest in therapeutic strategies generated to enhance innate immune sensing.

Activating the immune system is central to strategies aiming to trigger an adaptive immune response, and the innate immune system plays a crucial role in shaping the subsequent adaptive immune response. Adaptive immunity, characterized by specificity and memory, develops over time in response to specific antigens. The communication between the innate and adaptive immune systems is essential for developing an effective and targeted response against pathogens or abnormal cells. Activation of the adaptive immune response has propelled the field of immuno-oncology by shifting the focus of the therapeutic strategies from antigen-based targeted therapy to more general mechanisms of T-cell activation such as antibodies against checkpoint inhibitors [5,20] and the development of chimeric antigen receptor (CAR) T-cells [21,22]. Monoclonal antibodies targeting cytotoxic T-lymphocyte-associated protein 4 (CTLA-4) or programmed cell death protein 1 (PD-1) or its associated programmed cell death ligand 1 (PD-L1) have since been approved as monotherapies or in combination with existing treatments for various cancer types [23]. Additionally, and more recently, genetic manipulation of T-cells to generate CAR T-cell therapies has resulted in the development of effective therapies for some hematological malignancies [24]. These achievements in the clinical setting have given rise to a T-cell-centered, adaptive immune response perspective [25,26], thus undervaluing the requirement of innate immune responses to develop a robust anti-cancer T-cell response [27]. It is necessary to remember that innate immunity, and also adaptive immune response, play a role in the effectiveness of conventional treatments for cancer. This tenet has prompted the development of immunotherapy and conventional therapy combinations to achieve higher response rates and enhanced overall cancer patient survival [28,29,30,31]. In essence, the innate immune system provides the groundwork for the adaptive immune response. By sensing the presence of PAMPs from pathogens or DAMPs from distressed or dying cells, the innate immune system sets off a cascade of events that guide the adaptive immune system in recognizing, targeting, and creating long-term immunity against specific threats. This collaboration between innate and adaptive immunity forms a sophisticated defense strategy, allowing the body to effectively combat a wide array of dangers, from infections to cancerous cells, while also establishing memory for future protection. Investigators are exploring numerous strategies to activate the innate immune response. We discuss some of these most prominent and encouraging approaches in the following sections.

## 2. Discussion

### 2.1. Activation of Toll-like Receptors

Toll-like receptors (TLRs) are important mediators of inflammatory pathways. TLR expression is observed in immune cells such as dendritic cells (DCs), monocytes, natural killer (NK) cells, macrophages, and T-cells [11]. TLR signaling may contribute to anti-tumor effects by driving inflammasome activation in the tumor microenvironment [32,33]. Stimulation of TLR signaling results in the maturation of antigen-presenting cells such as DCs and macrophages, resulting in the production of type I IFNs and upregulation of surface co-stimulatory molecules such as CD86, CD80, and CD40 [11]. Because they can mediate the anti-tumor effect, TLR agonists such as imiquimod, which activates TLR7/8 pathways, have been approved by the U.S. Food and Drug Administration (FDA) to treat patients with basal cell carcinomas [34]. Interestingly, the Bacillus Calmette–Guerin vaccine is approved for the treatment of bladder cancer [35], where stimulation of TLR2 and TLR4 are responsible for the observed anti-cancer effect. A novel class of synthetic molecule TLR agonists, closely mimicking a pathogen-induced state, are injected intratumorally to enhance bioavailability and avoid toxic inflammatory syndromes associated with systemic delivery [36]. This strategy does not require prior identification of tumor antigens but instead relies on the endogenous antigen repertoire in the tumor microenvironment. The use of synthetic agonists targeting TLRs or other pathways that are involved in the recognition of PAMPs or DAMPs, such as stimulator of interferon genes (STING) and retinoic acid-inducible gene I (RIG-I)-like receptors, has shown encouraging results in preclinical models. The efficacy is further enhanced in combination with immune checkpoint inhibitors and regulatory T-cell depletion [37]. Following treatment, an upregulation of proinflammatory cytokines and tumor-specific effector T-cells has been observed in preclinical models [38].

Cytokines such as Fms-related tyrosine kinase 3 ligand (FLT3L) or granulocyte-macrophage colony-stimulating factor (GM-CSF) function to induce the commitment of hematopoietic progenitors to the DC lineage while also promoting enhanced DC proliferation and survival. FLT3L expression in preclinical models resulted, by acting through its cognate receptor, in the expansion of DC populations, which results in an amplified response to TLR3 agonists [39]. The enhanced TLR3 response stimulates an amplified IFN type I response, resulting in a significant lymphocyte infiltration, shifting the ratio to favor CD8+ effector cells over immunosuppressive CD8+ T-regulatory (Treg) cells within the tumor microenvironment. Human studies using combination therapy of FLT3L, TLR3 agonists, and radiation therapy have indicated that the additional DC activation and expansion leads to CD8+ cell activation and improved responses to immune checkpoint inhibitor therapy [40].

In the context of brain tumors, although resident antigen-presenting microglia express TLRs, they are unable to secrete interleukin (IL)-1B, IL-6, or tumor necrosis factor-alpha (TNF-α) to mount an effective innate immune response [41]. Attempts have been made to activate TLR signaling in preclinical models, including patient-derived GBM tumor cells. Researchers have successfully induced innate immune activation via melanoma differentiation-associated protein 5 (MDA-5) and RIG-I using cytosolic polyinosinic/polycytidylic acid (poly I:C) and 50-triphosphate RNA (3pRNA), respectfully [42].

The ample spectrum of TLR interactions provides a rationale for targeting these molecules for glioma therapy. In fact, TLR-based therapy in patients with gliomas has shown encouraging results as a single therapy or in combination with other therapies, such as chemotherapy, radiotherapy, or checkpoint inhibitors (reviewed in [43]).

### 2.2. Treatments That Induce a Stress Response

Treatments that induce a stress response, such as chemotherapy or radiation therapy, may cause cancer cells to become immunogenic as they release DAMPs and induce immunogenic cell death. Immunogenic cell death is a feature well described in preclinical models to promote phagocytosis in antigen-presenting cells [44], resulting in the secretion of immunomodulatory cytokines that, in turn, stimulate the recruitment of innate and adaptive immune cells to the tumor. In the context of GBM, current standard-of-care treatments such as temozolomide, radiotherapy, and chemotherapy all cause cell stress that can function to initiate immune responses in addition to direct tumor cell killing [45]. Although tumor cells themselves often fail to elicit a type I IFN response after radiation, dsDNA released from dying tumor cells is sufficient to activate cGAS-mediated type I IFN pathways in DCs [46]. In fact, tumor shrinking after radiation treatment was shown to be dependent on functional STING signaling in host cells, a central regulator of the innate immune response via activation of type I IFNs. These studies have led to significant interest in methods that induce dsDNA innate immune sensing in cancer cells. For instance, bromodomain and extra-terminal protein inhibitors have been shown to induce stress ligand expression in cancer cells. Expression of these surface molecules can activate an anti-tumor immune response because they can be recognized by NK cells, therefore triggering NK-mediated direct cancer cytolysis [47].

STING activation as a therapeutic tool for GBM treatment has recently been explored in preclinical murine and canine models with encouraging tumor responses that might be related to STING-mediated remodeling of the tumor microenvironment [48,49].

### 2.3. Strategies to Amplify the Innate Immune Response

The innate immune responses play a role in promoting CD8+ T-cell activation and function. Consequently, several strategies have been generated to enhance this effect. Studies using innate immune-stimulating cytokines, including granulocyte-macrophage colony-stimulating factor (GM-CSF), interleukins 2 and 15, type I IFNs, and FLT3L, are currently being explored to enhance antigen-presenting cell maturation and activation of NK cells.

GM-CSF enhances DC recruitment and differentiation; therefore, potential cancer vaccines (such as GVAX) using irradiated allogeneic cancer cells engineered to express GM-CSF are being tested [50]. In a different strategy to approach cancer treatment using GM-CSF, investigators designed talimogene laherparepvec (T-VEC), an oncolytic *herpes virus* engineered to express GM-CSF, which was approved by the U.S. FDA in 2015 for use in recurrent metastatic melanoma [51]. Further research is required to understand whether the GM-CSF approaches induce neoantigen-specific T-cell responses or if they simply amplify the existing T-cells in the tumor microenvironment.

To take advantage of the anti-tumor effects of FLT3L, investigators have designed adenoviral vectors to transfer FLT3L to cancer cells. After their success in preclinical models showing that FLT3L delivered by *adenovirus* vectors is sufficient to induce DC activation and prolong overall survival [52], a phase I clinical trial to test the safety of the combination of two adenoviral vectors expressing FLT3L and *herpes simplex virus* type 1 thymidine kinase (HSV1-TK) in patients with high-grade gliomas showed that the strategy is well tolerated and resulted in encouraging results [53].

### 2.4. Type I Interferons

Type I IFNs are the main players of immune responses in tumor and tumor microenvironment cells, facilitating efficient antigen presentation and myeloid cell migration to lymph nodes [54]. In response to type I IFN exposure, NK cells produce chemo-attractants such as chemokine (C-X-C motif), ligand (CXCL) 9, and CXCL10, influencing attraction to the tumor microenvironment of innate immunity-related cells [Wennerberg 2015]. When type I IFN is dysregulated in mouse models, NK cells show impaired functions, with decreased cytotoxicity and tumor infiltration [55]. Additionally, the type I IFN response is partly responsible for the efficacy of TLR vaccine strategies [56,57] and PD-1 checkpoint inhibition treatment [58]. Preclinical models have also connected type I IFNs with adaptive immune responses because they are essential during the priming of CD8+ T-cells [59,60].

Thus far, delivery of type I IFNs as antibody–cytokine conjugates has not yielded the anticipated clinical benefit, possibly due to several factors [61]. Type I IFN receptors are broadly expressed on cells, allowing for non-specific cell uptake and unpredictable toxicity. To overcome this limitation, low-affinity type I IFNs (IFNAR1) have been developed to interact more specifically with antigen-presenting cells. These low-affinity type I IFNs have demonstrated moderate anti-tumor activity with reduced toxicity [62].

Another interesting aspect of the STING/IFN pathway is related to the capability of this pathway to generate anti-tumor immunity by detecting tumoral DNA released by cancer cells. This DNA accesses the cytoplasm of DCs, where it is recognized by cGAS, with the subsequent activation of STING and IFN I production. Of interest is that this pathway might be causing the cross-priming of CD8+ T-cells against tumor antigens [63].

The solid data implicating IFN type I in the innate immunity against GBMs have been the basis of clinical studies. Specifically, IFN alpha-based therapy has been tested in combination with adjuvant temozolomide in a Phase III clinical trial for patients with newly diagnosed high-grade gliomas. The authors concluded that, compared with the standard regimen, temozolomide plus IFN alpha treatment could prolong the survival time of these patients. These results were more notable among patients with O6-methylguanine-DNA methyltransferase (MGMT) promoter unmethylation [64].

### 2.5. Direct and Indirect Activation of Innate Immunity Using Therapeutic Antibodies

Innate immune cells take advantage of an indirect interaction using Fc receptors (FcRs) to mount anti-tumor responses. These receptors bind to antibodies targeting tumor antigens and thus provide specificity for innate immunity-mediated tumor destruction. Therefore, therapeutic antibodies targeting tumor cells can be bolstered by various innate immune cell responses, such as antibody-dependent cell-mediated cytotoxicity by NK cells (ADCC) [65]. In other instances, after binding to antibodies during targeted therapy, innate immune cells containing FcRs with the immunoreceptor tyrosine-based activation motif (ITAM) induce cancer cell phagocytosis, enhancing anti-tumor responses [66].

Current strategies used in targeted antibody therapy utilize immunoglobulin (IgG)1 antibodies, which are known to bind a variety of FcγR commonly present in NK cells, monocytes, macrophages, and DCs, whose density and function are more tissue-specific. Preclinical models have shown that the innate immune cell effector functions assist in controlling tumor growth when using targeting monoclonal antibodies [67,68]. Approved antibody therapies such as trastuzumab, targeting human epidermal growth factor 2 (HER2/ERBB2), function not only by interfering with cell growth and signaling in cancer cells but also by inducing various innate immune effector processes [69]. Cetuximab, an approved monoclonal antibody to target epidermal growth factor receptor (EGFR) in colorectal cancer, has been shown to upregulate phagocytosis of EGFR overexpressing cancer cells, leading to enhanced T-cell-mediated anti-tumor immune response [70]. This collateral immune effect of anti-cancer receptor antibodies can be leveraged by combination with other treatments, such as chemotherapy, to induce stress responses, thus further increasing phagocytosis by macrophages and DCs [71]. Other monoclonal antibody therapies such as rituximab, which have shown similar anti-cancer immune responses in preclinical models, function by upregulating type I IFNs, which leads to maturation of DCs to bolster CD8+ T-cell effector functions [72]. In other studies, anti-CD47 antibodies specifically induced DC activation to cross-prime T-cell responses, with a limited effect on macrophages or other innate immune cells in syngeneic mouse models [28,73]. Signal-regulatory protein-α (SIRPα) is an inhibitory receptor containing an immunoreceptor tyrosine-based inhibitory motif (ITIM) that is present on various myeloid cells such as macrophages, DCs, and neutrophils. SIRPα functions to recognize overexpressed CD47 on the cell surface of tumor cells [74]. This interaction prevents cancer cells from being phagocytosed, resulting in an immunosuppressive tumor microenvironment. Disrupting the CD47–SIRPα interaction using anti-CD47 antibodies has been shown to enhance the phagocytic capability of macrophages [75]. In this direction, a phase I trial testing anti-CD47 antibodies has shown promise for the treatment of advanced solid and liquid tumors while demonstrating low toxicity [76]. CD47–SIRPα myeloid checkpoint inhibition has been shown to be effective in enhancing tumor phagocytosis and reducing tumor size in GBM studies [77,78]. A humanized monoclonal antibody targeting CD47, which directly hinders the CD47−SIRPα interaction, is currently undergoing clinical trials [76]. To address safety concerns, this antibody has typically been engineered on a human IgG4 framework to minimize Fc-dependent effector functions of the innate immune system. Similarly, an IgG1 version of the anti-CD47 antibody is expected to possess antibody-dependent cell-mediated cytotoxicity (ADCC) and antibody-dependent cellular phagocytosis (ADCP) activity against GBM. However, infusion-related toxicities and the obstacle of penetrating the blood–brain barrier currently restrict the systemic treatment of GBM with an IgG1 form of the anti-CD47 antibody.

Furthermore, a recent phase III clinical trial evaluated the efficacy of the combination treatment of temozolomide and the antibody-drug conjugate depatuxizumab mafodotin (ABT-414)—an EGFR antibody linked to monomethyl auristatin F—in newly diagnosed GBM [79]. The combination resulted in promising results in a phase II clinical trial that tested ABT-414 as a monotherapy and combined with temozolomide in adults and pediatric populations [80].

One limitation of the monoclonal antibody approach might be mediated by the presence of Fc polymorphisms. In fact, some of these polymorphisms expressed by innate immune cells interfere with the affinity of monoclonal antibodies, influencing the response to these therapies [81].

### 2.6. Innate Immune Checkpoint Therapy

Immune checkpoint therapy has revolutionized the field of cancer therapy. Immune checkpoint inhibitors such as ipilimumab (anti-CTLA-4 antibody) or pembrolizumab (anti-PD-1) have attracted the most attention due to their application to a wide variety of cancer types [82,83,84], with substantial benefit to 20% of treated patients [85,86]. Commonalities are seen between immunotherapy and conventional treatments, as they both increase the immunogenicity of the tumor and result in the release of DAMPS and tumor-associated antigens [87]. Innate immune systems can then amplify these signals through proinflammatory cytokines and chemokines to stimulate further cross-priming and recruitment of tumor-specific T-cells.

After successful clinical trials, several cancer vaccines have been approved for use in clinics [88]. Proinflammatory cytokines such as interleukin-2 and IFN-α trigger potent host innate immune responses, which, in turn, induce cross-priming and recruitment of T-cells to invoke an anti-cancer effect [89]. Expanding the range of immune checkpoint blockade has led to the development of broad-spectrum checkpoint inhibitors that target not only T-cells but also myeloid cell lineages in combination with or instead of T-cells. It seems reasonable to think that the simultaneous removal of the inhibitory pathways in innate and adaptive immune systems may eventually enhance T-cell functionality.

T-cell immunoglobulin and mucin-domain-containing-3 (TIM-3) is an example of surface receptor that regulates immune responses shared between innate immunity and adaptive immunity cell mediators, such as myeloid cells and T-cells. Thus, TIM-3 is closely associated with both immunosuppressive tumor-associated macrophages and T-cell exhaustion [90]. Importantly, TIM-3 impairs DC recognition of tumors by dysregulating nucleic acid sensing pathways [91]. Preclinical models have shown the anti-cancer effects of TIM-3 blockade in several tumor types [92]. Another immune checkpoint regulator is lymphocyte activation gene-3 (LAG-3), initially identified as an inhibitory receptor responsible for suppressing cytokine release and T-cell activation [93]. As with PDL-1 and TIM-3, the LAG-3 receptor is also present in myeloid and lymphoid cells, including NK cells and DCs and CD4+ T-cells, regulatory T-cells, and CD8+ tumor-infiltrating cells. Various mechanisms of blocking LAG-3 function are currently being assessed in clinical trials, including relatlimab, a monoclonal antibody used to treat patients with metastatic or unresectable melanoma [94].

### 2.7. CAR-NK Cells

NKs were first identified to kill tumor cells non-specifically [95,96]. Several anti-tumor effects have been demonstrated by NKs, functioning much like CD8+ cytotoxic T-cells without an antigen-specific T-cell receptor [97]. NK cytotoxicity is mediated through activation signals and the downregulation of certain human leukocyte antigens (HLA) on the cell surface of target cells. A review of the preclinical and clinical use of NK cells for brain tumor therapy has been recently reported by Fares and colleagues [98]. NKs can also bind the Fc region of antibodies bound to the cell surface of tumor cells, resulting in antibody-dependent cell-mediated cytotoxicity [58]. Studies have demonstrated the anti-glioma activity of EGFRvIII-specific CAR-T cells [99,100]; however, patients with GBM often solely express wild-type EGFR, limiting therapeutic benefits [101]. It has since been proposed to include CAR-T cells targeting both EGFR forms. Alternatively, CAR-NKs are suitable for treating GBM as they present minimal risk of affecting normal host cells [98,100]. Still, thus far, CAR-NKs, which have shown promising results in liquid tumors, are largely unexplored in the context of GBM.

NK cell lectin-like receptor subfamily C member 1 (NKG2A) functions as an inhibitory receptor found on both NK cells and T-cells, with a specific affinity for HLA-E, in contrast to classical major histocompatibility complex (MHC) class I molecules. HLA-E has been reported to be upregulated in many tumor types and promotes NK cell self-tolerance, leading to tumor evasion [102,103]. Therefore, NKG2A is a suitable target for anti-cancer strategies. In this regard, an IgG4-blocking monoclonal antibody, termed monalizumab, has been developed to block NKG2A functions and is currently being used as a stand-alone treatment or alongside anti-PD-1 therapy. Preliminary studies have already reported an improved NK cell and CD8+ T-cell anti-tumor response after NKG2A blockade [104,105].

### 2.8. Oncolytic Virotherapy

Oncolytic viruses (OVs) are ideal candidates to enhance patient responsiveness to immune checkpoint blockade. This is facilitated by the OVs’ unique ability to replicate and engage in anti-tumor activities using several distinct mechanisms. These mechanisms encompass a range of actions, including direct tumor cell destruction, modulation of the tumor microenvironment, recruitment of tumor-infiltrating lymphocytes, priming of immune responses driven by CD8+ T-cells and innate immune cells, as well as vascular alterations such as inhibiting tumor angiogenesis and neovascularization [106,107]. OVs can also induce immunogenic cell death of glioma cells [108], triggering the release of DAMPS that further stimulate innate immune functions, such as the recruitment and maturation of tumor-specific T-cells within the tumor microenvironment (Figure 2).

Recently, OVs have been engineered to serve as in vivo therapeutic delivery systems, taking advantage of their tumor cell selectivity. Arming OVs with transgenes that enhance existing methods of immune activation or novel immune modulatory mechanisms may improve patient response rate, allowing effective treatment to reach more patients [109,110,111,112]. This is particularly relevant in immunologically “cold” tumors as OVs have shown substantial T-cell infiltration in solid tumors, including GBM, which is often considered an immune desert [108]. Clinical trials have indicated that OVs are well tolerated and have resulted in encouraging results [108,113,114], providing an attractive opportunity for combination therapies using anti-cancer agents with complementary mechanisms of action.

Several OV families, including *adenoviruses*, *reoviruses*, *picornaviruses*, and *rhabdoviruses*, have been shown to induce robust innate immune activation through the protein kinase R pathway after treatment [115,116,117]. *Herpes simplex virus* has also been shown to induce type I and II IFNs, although utilizing an alternative pathway [118]. Innate immune activation is a double-edged sword in the context of OV immunotherapy, as it is required for effective lymphocyte recruitment but, in turn, hinders virus replication [119]. Preclinical studies using engineered OVs to induce innate immune responses have shown promise and may become an important component of immune checkpoint blockade, particularly in immune “cold” malignancies. This potential combination of OVs and immune checkpoint inhibition is underscored by the current clinical development of immune checkpoint inhibitors in combination with TLR agonists like STING and RIG-I [120,121]. Recent preclinical studies have incorporated OV and CAR-T therapy [122,123,124,125,126], following the basis that the OV-mediated inflammation and direct cell killing will induce antigen expression and antigen spreading, therefore promoting migration and proliferation of T-cells to the tumor microenvironment.

Additionally, OVs offer a promising route to avoid unacceptable toxicity associated with systemic delivery and can instead take advantage of tumor-selective replication to express anti-tumor molecules [119,120]. This may circumvent, among other obstacles, the current roadblocks observed when using treatments aimed at inducing type I IFN responses.

### 2.9. Manipulation of Microbiome and Innate Immunity against Cancer

Microbiota is actively being studied in the context of cancer therapeutics, with particular interest in immune modulation [127,128]. In preclinical models, it has been reported that responses to immune checkpoint blockade targeting PD-1/PD-L1 and CTLA-4 are influenced by host microbiota [129,130,131]. Given the role of the innate immune system in bacterial detection and tolerance and the expression of PD-L1 on innate immune cells, it is reasonable to expect that microbiota contributes towards regulating innate immunity [132]. The link between microbiota and DC activation and maturation has also been shown in germ-free mice supplemented with *Bacteroides fragilis*, resulting in improved responses to CTLA-4 inhibition [133]. Recently, it has been proposed that gut microbiota may be a predictive marker of the response to immunotherapy, where *Akkermansia muciniphila* and *Enterococcus hirae* were shown to upregulate IL-12 in DCs to facilitate an effective immune response during normal microbiota conditions [134]. Furthermore, studies have implicated microbial metabolites such as short-chain fatty acids or tryptophan derivatives as factors involved in patient response to immunotherapy [131]. In the clinical setting, microbiota diversity has been linked to NK cell abundance in peripheral blood and improved patient response to immunotherapy [135]. The advancement of sequencing technologies, including metagenomic sequencing, facilitates the exploration of the microbiome in the context of diseases, including cancer. Delineating which innate immune cells specifically respond to microbiota cues and which are the molecular underpinnings of these functional relationships may provide new avenues to develop anti-cancer therapies.

## 3. Conclusions

The immune response to cancer consists of a complex chain of events that initially involves the innate immune response followed by the adaptive immune response. In cancer immunotherapy, innate immune signaling pathways are coerced to aid in tumor evasion and immune surveillance. Consequently, mechanisms required to activate these pathways are likely essential for cancer treatment strategies. Importantly, innate immune activation has been shown to play an important role in priming the adaptive immune response against cancer. Emerging evidence suggests that modulation of specific innate immune pathways may provide benefits as a monotherapy or in combination with existing approaches.

The evolution of therapeutic strategies for gliomas involves a multifaceted approach aimed at activating the innate immune system to combat these challenging brain tumors. Novel approaches include activating Toll-like receptors to trigger an immune response against the tumor, harnessing treatments inducing stress responses within the cancer cells, and employing methods to bolster the innate immune response for enhanced tumor recognition and elimination. Furthermore, utilizing IFN type-I therapy, therapeutic antibodies, and immune checkpoint antibodies can help modulate the immune system response, augmenting its ability to effectively target and destroy glioma cells. Leveraging the cytotoxic capabilities of NK cells, oncolytic virotherapy, and manipulating the microbiome within the body offer promising avenues to stimulate the innate immune system, either directly targeting the tumor or creating a favorable environment for an intensified immune response against gliomas. The ongoing exploration and integration of these diverse approaches hold substantial promise for developing future glioma therapies that harness the power of systemic innate immunity to improve treatment outcomes and patient prognosis.

To conclude, we can speculate on the possibility of combining innate immunity activation with conventional treatments such as chemotherapy and radiotherapy. In an ideal scenario, these combinations would result in specific biological interactions that would cause synergistic effects. Alkylating agents, including temozolomide, induce DNA damage in rapidly dividing tumor cells. When coupled with TLR activation, the latter triggers downstream signaling pathways, for instance, NF-κB and mitogen-activated protein kinase (MAPK), leading to the production of proinflammatory cytokines. TLR activation, particularly in tumor-associated macrophages and dendritic cells, also induces a proinflammatory microenvironment. This environment is characterized by the release of danger signals, which eventually increase the presentation of tumor antigens, thus favoring the immune system recognition of the glioma cells. In the case of radiotherapy and other DNA damage treatments, the induction of immunogenic cell death will promote the release of tumor antigens and DAMPs such as high mobility group box 1 (HMGB1) and adenosine triphosphate (ATP), triggering the innate immune reaction. Thus, the DAMPs released due to radiation-induced immunogenic cell death act as chemoattractants and pave the way for the subsequent activation and recruitment of innate immune cells, including NK cells. NK cells can, in turn, recognize and eliminate stressed or malignant cells, complementing the effects of radiotherapy by targeting surviving or radioresistant tumor cells. One of the mechanisms proposed to support the abscopal effect triggered by radiation therapy involves the systemic activation of immune cells. Thus, activation of innate immune response in combination with radiotherapy can further potentiate this effect, leading to a wider-reaching immune response against glioma cells both within and outside the radiation field. In summary, the rationale behind the combinations of innate immunity activation and conventional treatment of gliomas and other cancers lies in their ability to create a synergistic effect, leading to the activation of the adaptive immune response to recognize, target, and eliminate glioma cells more effectively. Combining these emerging immunotherapeutic strategies with established treatments like radiotherapy and chemotherapy opens exciting avenues to improve treatment efficacy and patient outcomes in glioma management.

## Figures and Tables

**Figure 1 ijms-25-00947-f001:**
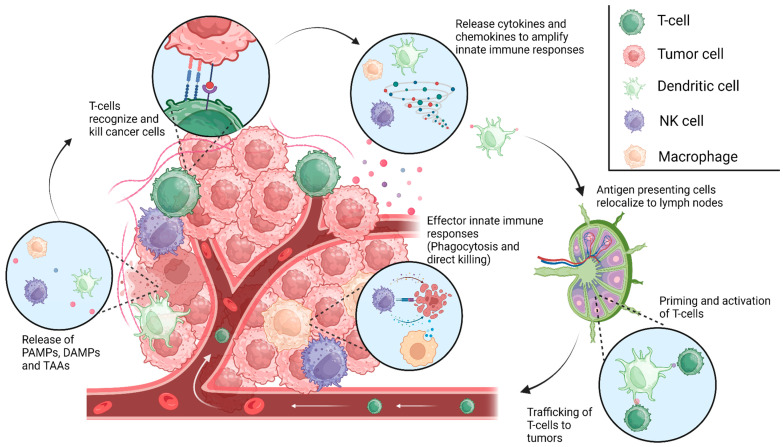
The cancer immunity cycle. Innate immune cells facilitate immune responses following the recognition of PAMPs, DAMPs, or other unique tumor-associated antigens (TAA), resulting in antigen presentation in the associated draining lymph nodes. This process allows for the priming and activation of T-cells and adaptive immunity, which aid in tumor elimination, producing more PAMPs, DAMPs, and TAAs to continue the cycle. Escape from these innate immune mechanisms might lead to the formation of gliomas and the malignant phenotype of these tumors. [Created with BioRender.com].

**Figure 2 ijms-25-00947-f002:**
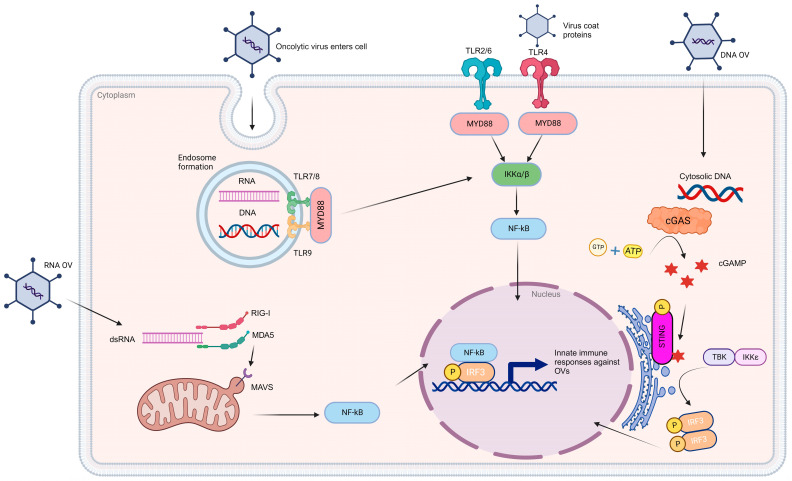
Oncolytic viroimmunotherapy induces innate immune responses. The delivery of oncolytic viruses (OVs) triggers a range of innate immune responses that are contingent on the species of origin. The detection of these viruses hinges on pattern recognition receptors, including Toll-like receptors (TLRs), retinoic acid-inducible gene I (RIG-I)-like receptors (RLRs), and nucleic acid sensors. TLR7/8 plays a key role in recognizing single-stranded ribonucleic acid (RNA), while TLR9 is responsible for identifying hypomethylated CpG deoxyribonucleic acid (DNA). Nevertheless, both of these pathways converge to activate the myeloid differentiation primary response 88 (MYD88) pathway, ultimately leading to the activation of nuclear factor kappa-light-chain-enhancer of activated B cells (NF-Kββ). Similarly, OVs detected through surface receptors such as TLR2/6 and TLR4 also initiate the MYD88 pathway. Within the cytosol, the presence of OV double-stranded RNA (dsRNA) is detected through the RIG-I and MDA5 receptors, which, in turn, activate mitochondrial antiviral-signaling protein (MAVS) located on the mitochondria. MAVS, in collaboration with the inhibitor of nuclear factor kappa-B kinase subunit epsilon (IKKε)/tank-binding kinase 1 (TBK1), leads to NF-Kβ-mediated immune activation. In the case of DNA-based OVs, their presence is recognized by cytosolic DNA sensors, particularly through the cGAS-STING pathway. OV DNA triggers enzymatic activation of cyclic GMP-AMP synthase (cGAS), which catalyzes the synthesis of cyclic GMP-AMP (cGAMP) using ATP and GTP as substrates. cGAMP subsequently binds to a stimulator of IFN genes (STING) and triggers the activation of the IKKε/TBK1 complex. This complex, in turn, phosphorylates IFN regulatory factor 3/7 (IRF3/7), leading to the expression of genes involved in the innate immune response. Once NF-Kβ and IRFs are activated, they translocate to the nucleus and induce the expression of proinflammatory cytokines, chemokines, and type I/III IFNs. These IFN-stimulated genes establish an antiviral state, which, in turn, plays a critical role in activating adaptive immunity. [Created with BioRender.com].

## Data Availability

All data related to this review is present in the manuscript.

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
