# Peer review of "Targeting Innate Immunity in Glioma Therapy"

_ijms, 2024, doi:10.3390/ijms25020947_

Round 1

Reviewer 1 Report

Comments and Suggestions for Authors

Title and Abstract:

  1. The title is clear and relevant to the content. However, it might be beneficial to specify the focus on glioma therapy in the title.
  2. The abstract provides a concise overview, but consider adding specific findings or contributions of the review to enhance clarity.
  3. Consider adding specific insights or findings mentioned in the review to make the abstract more informative.
  4. Advancing the treatment of GBM requires a deeper understanding" -> "Advancing GBM treatment requires a deeper understanding..."
  5. Consider rephrasing: "Our review provides a brief overview of innate immunity, followed by a discussion of current therapies aimed at boosting the innate arm of the immune system."
  6. Specify what the "innate arm" refers to in the immune system.

Introduction:

  1. The introduction effectively establishes the context, emphasizing the need for improved therapies for glioblastomas (GBMs).
  2. The challenges in GBM treatment are well-described, including issues of early infiltration and heterogeneity.
  3. Consider providing more recent statistics or developments in GBM research or treatment if available.
  4. The overview of innate immunity is well-written, providing a foundational understanding for readers.
  5. Consider simplifying complex sentences for improved readability.
  6. Use more recent references for foundational concepts in innate immunity if available.
  7. The section discussing new aspects of innate immunity in tumorigenesis is informative.
  8. The feedback loop involving DAMPs, tumor-specific CD8+ T-cells, and therapeutic implications is well-presented.
  9. Consider breaking down complex concepts into smaller sentences for easier comprehension.
  10. The connection between innate and adaptive immunity is explained well, emphasizing their collaborative role in the immune response.
  11. The discussion on recent developments in immuno-oncology is insightful.
  12. Clarify or elaborate on the mention of innate immunity's role in the effectiveness of conventional cancer treatments.
  13. The transition to discussing strategies for activating the innate immune response is smooth.
  14. Each strategy is introduced well, but consider providing brief definitions or explanations for terms such as PAMPs and DAMPs to assist readers less familiar with immunology.
  15. Provide references for statistics or information on GBM treatment if available.
  16. Clarify or provide a reference for the term "immune cold" regarding GBM microenvironment.
  17. Specify the figure number for Figure 1 when referencing it in the text.
  18. Ensure that the references for PAMPs and DAMPs are recent and widely recognized.
  19. Consider providing more recent references (if available) for the statements on abnormal cell proliferation and stress associated with carcinogenesis.
  20. Provide references for recent developments in immuno-oncology mentioned in the text.
  21. Elaborate on the mention of innate immunity's role in the effectiveness of conventional cancer treatments.
  22. Consider adding brief explanations or definitions for terms such as PAMPs and DAMPs to aid readers less familiar with immunology.
  23. "GBM diagnosis is associated with a median survival of approximately 15 months." -> "GBM diagnosis is associated with a median survival of approximately 15 months, and these tumors are known for early infiltration, limiting the effectiveness of surgical resection."
  24. Consider clarifying or providing references for terms like "immune cold" and "desert."
  25. "PAMPs (Pathogen-Associated Molecular Patterns) and DAMPs (Damage-Associated Molecular Patterns)" -> Consider providing this as a parenthetical explanation after the first mention.
  26. "This feedback loop, in turn, facilitates the recruitment, activation, and clonal expansion of tumor-specific CD8+ T-cells, which may promote response to therapy and patient outcome." -> Break this down into smaller sentences for clarity.
  27. "Activation of the adaptive immune response has propelled the field of immuno-oncology, which has resurfaced based on a paradigm shift that has shifted the focus..." -> Consider rephrasing for clarity.
  28. "We discuss some of the most prominent and encouraging in the following sections." -> Specify what is discussed in the following sections.

Overview of Innate Immunity:

  1. Consider including a brief mention of the blood-brain barrier's role in limiting immune cell infiltration in GBM, which contributes to its immune-cold nature.

Activation of Toll-Like Receptors:

  1. Clarify the role of STING (stimulator of interferon genes) in the context of recognizing cyclic dinucleotides and its involvement in the recognition of DNA viruses.

Treatments that Induce a Stress Response:

  1. Specify the types of stress responses induced by chemotherapy or radiation therapy and their relevance to immunogenic cell death.

Strategies to Amplify the Innate Immune Response:

  1. Provide more context on how FLT3L contributes to DC activation and why enhanced TLR3 response leads to significant lymphocyte infiltration.

Type I Interferons:

  1. Mention the importance of STING in the context of type I interferon response induced by DNA released from dying tumor cells.

Direct and Indirect Activation of Innate Immunity Using Therapeutic Antibodies:

  1. Emphasize the significance of Fc receptor polymorphisms in influencing the efficacy of therapeutic antibodies and their interactions with innate immune cells.

CAR-NK Cells:

  1. Provide more information on the advantages and challenges associated with CAR-NK cells, especially in comparison to CAR-T cells.

Manipulation of Microbiome and Innate Immunity Against Cancer:

  1. Include examples of specific microbial metabolites and their roles in patient responses to immunotherapy.

Conclusions:

  1. The conclusions section is well-structured, summarizing key points. Consider reiterating the potential challenges or limitations of the discussed strategies.

Author Response

We thank the reviewers for their comments and suggestions. Please find a point-by-point response in the attached document.

Sincerely,

Candelaria Gomez-Manzano, MD, FAAN

Reviewer 2 Report

Comments and Suggestions for Authors

The proposed review aims to illustrate the current state of the art regarding the therapy of gliomas in general and glioblastoma.

The topic is very actual as glioblastoma is a lethal and widespread brain tumor for which there is neither prevention nor effective therapeutic strategies.

In my opinion, the proposed review fails to develop what is proposed.

In fact, the structure of the review describes in a relatively in-depth way the molecular functioning of innate immunity but neglects many specific aspects related to the subject.

Especially:

1) The pathogenesis of gliomas and their relationship to innate immunity is not developed

2) The text neglects the important differences between glioblastoma and gliomas

3) In the introductory part, the relationship between the effectors of innate immunity or in general and the blood-brain barrier, glial cells, inflammation and more is not adequately developed

4) In my opinion, preclinical and clinical studies are insufficiently described and the review should do so in a more in-depth way, perhaps even discussing them critically.

5) The review should be accompanied by tables with references to preclinical studies and ongoing clinical trials and trials

Author Response

We thank the reviewer for their comments and suggestions. Please find a point-by-point response in the attached document.

Sincerely,

Candelaria Gomez-Manzano, MD, FAAN

Round 2

Reviewer 1 Report

Comments and Suggestions for Authors

 Accept in present form

Reviewer 2 Report

Comments and Suggestions for Authors After a careful and accurate reading of the proposed new manuscript, I confirm my doubts about the proposed manuscript. The review speaks only marginally about the state of the art of glioblastoma immunotherapy and instead favors a long and in-depth illustration of some aspects relating to innate immunity and the pathogenesis of oncological diseases in general. Most of the relevant clinical studies are neither cited nor considered. I consider the setup problems detected to be unresolved. Most of the topics covered are off topic.